# Diabetes Mellitus Is a Chronic Disease that Can Benefit from Therapy with Induced Pluripotent Stem Cells

**DOI:** 10.3390/ijms21228685

**Published:** 2020-11-18

**Authors:** Felipe Arroyave, Diana Montaño, Fernando Lizcano

**Affiliations:** 1Doctoral Program in Biosciences, Universidad de La Sabana, Chía 250008, CU, Colombia; felipe.arroyave@unisabana.edu.co; 2Center of Biomedical Investigation (CIBUS), Universidad de La Sabana, Chía 250008, CU, Colombia; diana.montano2@unisabana.edu.co

**Keywords:** regenerative medicine, iPSC, diabetes mellitus, pancreatic β-cells, transcriptional regulation, histone modification

## Abstract

Diabetes mellitus (DM) is one of the main causes of morbidity and mortality, with an increasing incidence worldwide. The impact of DM on public health in developing countries has triggered alarm due to the exaggerated costs of the treatment and monitoring of patients with this disease. Considerable efforts have been made to try to prevent the onset and reduce the complications of DM. However, because insulin-producing pancreatic β-cells progressively deteriorate, many people must receive insulin through subcutaneous injection. Additionally, current therapies do not have consistent results regarding the prevention of chronic complications. Leveraging the approval of real-time continuous glucose monitors and sophisticated algorithms that partially automate insulin infusion pumps has improved glycemic control, decreasing the burden of diabetes management. However, these advances are facing physiologic barriers. New findings in molecular and cellular biology have produced an extraordinary advancement in tissue development for the treatment of DM. Obtaining pancreatic β-cells from somatic cells is a great resource that currently exists for patients with DM. Although this therapeutic option has great prospects for patients, some challenges remain for this therapeutic plan to be used clinically. The purpose of this review is to describe the new techniques in cell biology and regenerative medicine as possible treatments for DM. In particular, this review highlights the origin of induced pluripotent cells (iPSCs) and how they have begun to emerge as a regenerative treatment that may mitigate the pathology of this disease.

## 1. Introduction

Diabetes mellitus (DM) is a chronic, noncommunicable disease with high morbidity and mortality due to chronic deterioration of insulin-producing cells. Diabetes triggers a series of vascular events that affect most of tissues and is the main cause of kidney failure, vision loss, ischemic heart disease, strokes, and peripheral artery occlusive disease [1,2]. DM is currently considered the seventh leading cause of death worldwide, and it was estimated in 2019 that a total of 9.3% of the world population suffered from this disease [3]. DM is closely related to the disruption of the body’s energy balance, which includes a sedentary life together with a high caloric intake that induces obesity. The presence of complications in DM has devastating implications, leading to a deterioration in the quality of life for those who suffer from the disease. DM is classified as type 1 DM, in which the insulin-producing beta cells (β-cells of the pancreatic islets) are destroyed because of an autoimmune response, and type 2 DM, in which a long period of alteration in the peripheral action of insulin causes progressive deterioration in the activity of β-cells in the pancreas [4]. Medical therapies for DM seek to establish good control of the level of glucose in the blood. For type 1 DM, the treatment is insulin replacement due to the total absence of this hormone in the body. Extensive effort has been taken to simulate the normal physiology of insulin after exogenous administration. However, because insulin is easily degraded in the digestive system, it must be administered parenterally. Despite significant advances, many obstacles, challenges, and doubts surround insulin administration [5]. Advances in disease treatment are more focused on establishing a pattern of strict control between levels of insulin in the blood and subcutaneous administration of insulin through the use of increasingly complex devices [6,7,8]. Human insulin was the first peptide hormone synthesized using the recombinant DNA technique in the early 1980s [9]. At present, some amino acids in this recombinant insulin have been modified to reestablish the physiological effect of endogenous insulin using short-, intermediate- or long-acting insulins [10,11,12]. The number of drugs to normalize glucose levels in type 2 DM patients is ostensibly increasing. These medications function to increase insulin activity in peripheral tissues, specifically muscle and fatty tissues [13,14,15]. Some mediate insulin secretion after glucose stimulation through the incretin effect [16,17], while others impede hepatic glucose production, and recently, antidiabetic drugs have been applied to increase the excretion of glucose in the urine [18,19]. For a few years, the government regulatory agencies that oversee drug production have indicated that a favorable effect on complications, especially cardiovascular risk and deterioration of kidney function, is a prerequisite for new medications for DM [20,21,22].

Although there are many systems that mitigate or delay the effects that DM can have on human health, the treatments that exist to counteract the complications of the disease have not yet achieved the desired level of success [23,24]. There is a growing consensus that a regenerative medicine approach is necessary for the treatment of DM both in cases of type 1 DM and in cases of irreversible damage to the pancreas in type 2 DM. For this reason, enhancing or inducing the intrinsic regenerative ability of endocrine islets and developing new strategies to produce insulin-secreting β-cells will have profound consequences for the development of therapeutic treatments for diabetes [25,26,27,28]. Islet transplantation has been attempted in a large population of diabetic patients, but because of the shortage of cadaveric pancreata, these attempts met a limited amount of success.

Therefore, there is consensus among physicians, biologists and chemists that one of the best proposed alternatives is regenerative medicine that involves β- cells from pancreatic islets derived from human pluripotent stem cells (hPSCs) [29]. In recent years, the development of knowledge related to the biology of hPSCs has allowed us to propose them as a viable alternative for many chronic noncommunicable diseases. hPSCs can be acquired from embryonic blastocysts, the umbilical cord and some cellular “niches” in adult tissue [30]. Furthermore, differentiated pancreatic β-cells can be reprogrammed from somatic cells-a fact that has opened new avenues in regenerative medicine. Because human induced pluripotent cells (hiPSCs) can be obtained from the same individual, issues related to ethical problems and rejection of the new tissue can be avoided [25,31].

## 2. DM and Public Health

Most hospital care is devoted to noncommunicable chronic diseases, and their complication-related expenses exceed the annual budgets of the public health systems of almost every country in the world [32,33]. Among noncommunicable chronic diseases, cardiovascular diseases cause the highest number of deaths. The main cardiovascular risk factors related to mortality are high blood pressure (13%), smoking (9%), DM (6%), overweight and obesity (5%) [34].

Type 2 DM is closely related to the progressive increase in a person’s weight, causing overweight and obesity. The chronic decline in glucose and lipid metabolism leads to vascular complications [4,35]. DM reduces life expectancy, and the implications of vascular complications in terms of quality of life are devastating for both the person with DM and their family [36]. The International Diabetes Federation (IDF) predicted that by 2019, one in 11 adults will have some type of diabetes, totaling almost 463 million people worldwide [37,38]. Moreover, one in two persons with type 2 DM are not properly diagnosed. Currently, 75% of people with DM live in countries with low- or moderate-income levels [39,40]. This could indicate drawbacks in public health systems that could lead to problems ranging from misdiagnosis to treatment failure [41,42].

Given the pathophysiology of complications in patients with DM, it is reasonable to argue that one of the major underlying issues for patients with DM is their increased cardiovascular risk due to chronic inflammation, impaired endothelial function, augmented free radicals, increased polyol pathways and glycation of proteins [43,44]. Consequently, there is an increased risk of atherogenesis [45]. DM is closely associated with atherosclerotic vascular disease, and more than 50% of type 2 DM patients already have early vascular complications at diagnosis [46,47,48].

In 2019, IDF estimated that total diabetes-related health expenditure reached 760 billion USD. These estimates may be low, as they consider that the expenses due to diabetes are constant [49,50]. The global prevalence of diabetes in adults over 18 years of age has increased from 4.7% in 1980 to 9.3% in 2019, especially in underdeveloped countries [51]. In 2016, approximately 1.6 million deaths were directly caused by diabetes, while close to 2.2 million deaths were attributable to a high glucose levels in the blood [50,52]. The number of people with prediabetes could exceed 700 million worldwide, many of which live in underdeveloped countries [53]. Three in four people living with type 2 DM are working age (aged 20–64 y), and this number is expected to increase to 486 million by 2045 (Figure 1).

There is clear evidence that the health conditions in the COVID-19 pandemic have worsened the circumstances of people with obesity and type 2 DM. Health care services and, in some cases, access to medicines and supplies, have been disrupted [54]. Diet and physical activity are pillars of diabetes self-management and can reduce the risk of poor outcomes in people with disabilities as well as those with cardiometabolic multimorbidities. The current pandemic and social isolation are likely to increase rates of obesity, anxiety, and depression, which can also lead to poorer medication adherence and less control over risk factors [55,56].

DM and the continual increase in blood glucose levels, even to levels below the diagnostic threshold, are associated with a wide range of cardiovascular conditions that collectively comprise the largest cause of both morbidity and mortality for people with DM [57]. Systematic reviews indicate that the relative risk of cardiovascular diseases (CVD) is between 1.6 and 2.6, but that the relative risk is higher among young people and slightly higher in women [58]. Among people with DM, especially those living in countries with moderate or low income levels, 21% (12–32%) suffer from coronary artery disease, and 33% develop cardiovascular disease [48,59]. Excess glucose is associated a 15% increase in all-cause mortality, which includes mortality from cardiovascular disease and kidney disease [60]. However, the relative risk of each patient is variable and depends on the region of origin of the patients [61,62,63]. DM can cause microvascular damage to the retina, kidney, or nerves. Diabetic retinopathy is one of the main causes of blindness in the working population and has devastating personal and economic effects. The risk of diabetic retinopathy increases with the course of the disease, and it is more common in patients with type 1 DM than in those with type 2 DM. In type 2 DM, there are obvious differences in incidence rates across races, ranging from 21% in Asians to 56% in African-Americans [64,65]. Diabetic nephropathy is a leading cause of end stage renal disease. Hyperglycemia induces hyperfiltration and morphological changes in the kidneys that ultimately lead to increased urinary albumin excretion (albuminuria), podocyte damage and loss of filtration surface [66,67,68]. Globally, more than 80% of end-stage renal disease is caused by diabetes, hypertension, or a combination of both. The proportion of end-stage renal disease cases attributed to diabetes varies between 10% and 67%. The prevalence of end-stage renal disease is also up to 10 times higher in people with DM than in those without DM [69,70]. Diabetic neuropathy is a complication of both type 1 and type 2 DM. The cause of neuropathy in diabetic patients is a matter of discussion, hyperglycemia is the main factor that induces neuropathy via polyol pathway hyperactivity along with a decrease in antioxidants, the regeneration of glutathione, an increase of advanced glycation end products (AGEs), and the activation of diacylglycerol and protein kinase-c isoforms [71]. Depletion of glutathione could be the primary cause of oxidative stress and could be related to the accumulation of toxic species [72]. Hyperglycemia leads to a reduction in peripheral perfusion, followed by ischemia of the peripheral nerves and alteration of the myelin sheath covering the nerves. Pain, dysautonomia and diabetic foot complications are the main consequences of neuropathy in diabetes [73]. The prevalence of diabetic neuropathy ranges from 7% within 1 year of diagnosis to 50% for those with diabetes for >25 years [74]. If patients with subclinical levels of neuropathic disturbances are included, the prevalence might exceed 90% [75].

The risk of suffering from DM and its complications depends on genetic variations influenced by environmental circumstances that affect pancreatic β-cells development and function and insulin secretion and sensitivity. In type 1 DM, the selective destruction of β-cells is caused by multifaceted interaction between risk-conferring genes and environmental factors. Predisposition is in part subject to the expression of human leukocyte antigen (HLA) genes. The loci with the highest susceptibility to type 1 DM are HLA-DRB1 and HLA-DQB1 on chromosome 6p21, which contributes about 50% of the genetic risk [35,76,77]. At present, 58 genomics regions show substantial evidence for type 1 DM association and about 50 genes are suggested to be potential causal of disease [78,79]. On the other hand, many SNPs variants of non-HLA observed in patients with diabetes mellitus are not causal, there is a growing interest in the deepening of knowledge of functional alteration of genes based on GC content, distance to the transcription start site, DNase peak or footprinting, histone modifications, transcription factors motifs and weight matrix suggest that at list these genes interleukin 27 (IL27), pro-apoptotic BH3-only (BAD), the regulator of inflammatory response CD69, protein kinase C theta, (PRKCQ), CLEC16A, ERBB3, and CTSH, may have a causal effect on the appearance of DM [80,81,82,83,84].

More than 95% of patients with type 2 DM have a genetic origin that conforms to a polygenic model. Approximately 70 loci were identified from the analysis of aggregated controls of 150,000 cases [85]. Taken together, these loci represent approximately 6% of the heritability for type 2 DM. Of these loci, a single-nucleotide polymorphism (SNP) in the transcription factor 7 like 2 (TCF7L2) (with the allele increasing risk present at a frequency of 30%) has the greatest overall effect on risk, conferring a 1.4-fold increase in risk per allele. However, for many of the variations associated with type 2 DM, the mechanism that contributes to disease risk is unknown [86,87].

## 3. Pancreas and *β*-Cell Development

An understanding of the development of regenerative medicine in DM requires knowledge of the embryonic development of the pancreas. Our comprehension of embryonic development in humans is largely subject to observations made in other mammals. The pancreas is both a digestive organ that participates in the digestion of macronutrients through the production of digestive enzymes and an endocrine organ that regulates metabolism of cells by controlling the intracellular transport of glucose. Digestive function is ensured by acinar cells that secrete digestive enzymes into the pancreatic ducts [88]. Pancreatic endocrine cells are clustered into islets of Langerhans, which are composed of different functional cells. The endocrine pancreas consists of 5 main types of secretory cells: alpha cells (α-cells), which express glucagon; β-cells, which express insulin; delta cells (δ-cells), which express somatostatin; gamma cells (γ-cells), which express pancreatic polypeptide; and epsilon cells (ε-cells), which express ghrelin. In adults, 60% of islet cells are β-cells, and 30% are α-cells. The different types of endocrine cells in the islets of Langerhans establish a paracrine network and interact with peptides involved in autocrine-mediated hormone secretion [89,90] (Figure 2).

Human embryonic development can be described as a 23-stage process according to that established by the Carnegie Institutes [92]. The development of the pancreas begins around stage 9 of Carnegie (CS9) from the anterior intestine of the primitive endoderm [93]. The specification of the pancreatic space is determined by signals that originate from the mesoderm, including transforming growth factor b (TGF-β), retinoic acid (RA) and fibroblast growth factor (FGF) [94,95]. Pancreatic growth factor and duodenal homeobox factor 1 (PDX1) are key to the initial development of the pancreas [96]. The first step is invagination of the foregut into dorsal and ventral buds, which later fuse to form the pancreas. The pancreatic buds are formed by stratified epithelium that stochastically polarizes, forming the micro lumens that later form the pancreatic ducts. The dorsal and ventral buds of the pancreas are associated with the appearance of the growth factors sex-determining region Y (SRY)-box 9 (SOX9), PDX1 and GATA binding protein 4 (GATA4), which are necessary for subsequent pancreatic growth (CS10-13) [97,98,99]. During the CS13 period, microlumens begin to appear in the dorsal bud, representing the first sign of the network in the exocrine tubules, which eventually drain into the intestine. In mice and humans, proliferation is dependent on signals from the mesenchyme and also from cell-to-cell interactions, notably via the NOTCH pathway, which activates the transcription factor HES1 [100].

Endocrine differentiation occurs from multipotent or bipotent endocrine-ductal progenitors and is marked by the expression of NEUROG3 factor (NGN3), which controls various transcription factors that determine the identity of cells in the islets of Langerhans [101]. NGN3 expression increases rapidly after the embryonic period, with the appearance of fetal α-cells at the beginning, followed by β-cells, which produce insulin and are the most predominant islet cell type in human development [102,103]. Upon a transient increase in the expression of NGN3, these progenitors stop proliferating and differentiate into endocrine cells. It has been observed that the expression of NGN3 is necessary for the commitment of progenitor cells to an endocrine fate, as NGN3-null mice are completely lack both intestinal and pancreatic endocrine lineages [93,104]. Although the results of studies on the factors that control the differentiation of different cell types in pancreatic islets are not conclusive, factors such as paired box protein pax 4 (PAX4), hepatocyte nuclear factor 4 alpha (HNF4α), hepatocyte nuclear factor 3-β (HNF3β/FOXA2), Nirenberg and Kim homeo box 6.1 (NKX6.1), motor neuron and pancreas homeobox-1 (MNX-1), V-maf musculoaponeurotic fibrosarcoma oncogene homolog A (MAFA) and PDX-1 have important roles in β-cell formation, differentiation and function [105,106,107,108,109,110,111]. Pdx-1 mRNA expression is downregulated, with expression restricted only to endocrine cells in the pancreas and preserved in adult β-cells. Pdx-1 contains three principal transcription initiation sites [112], and in β-cells, each of these sites may be activated by the binding of a specific site of transcription factors [113]. Tissues such as the pancreas and liver lack the ability to regenerate because endocrine progenitors or pancreatic adult stem cells do not exist in the mature tissues. Thus, β-cells regeneration depends mainly on β-cell replication, which declines with aging [114,115].

## 4. Current Treatments of Regenerative Medicine for DM

A healthy adult has a blood volume of 5 L containing approximately 5 g of glucose. Maintaining blood glucose levels is critical to meet the energy demands of the body’s tissues, specifically the brain. Despite fluctuations in glucose consumption and caloric expenditure, blood glucose levels are strictly regulated within a few mg/dL, by the precise secretion of hormones produced in the pancreatic islets of Langerhans [116]. Insulin has been the indispensable treatment for both patients with type 1 DM and those with type 2 DM in which the endogenous production of insulin by the pancreas is exhausted. The complexity of DM is such that control of blood glucose levels alone is not sufficient to reduce the occurrence of cardiovascular complications [117]. This is, in part, because a series of changes in inflammatory, metabolic and redox conditions are involved in the onset of microvascular complications (kidney failure, blindness, etc.) and macrovascular complications (myocardial infarction, peripheral vascular disease and strokes), which decrease the quality of life of patients [118,119,120].

The therapies that exist to counteract the effects of DM have not yet managed to offer a definitive cure. Transplant of the pancreas has been used since the beginning of this century on a regular basis [121,122]. However, the morbidity of the procedure or the difficulties of the technique limit its use.

Moreover, pancreatic islet transplantation has emerged as complementary to pancreas transplantation with very good results. Islets transplants has emphasis as an alternative to insulin therapy. Thus, should be focused on achieve an excellent glycemic control without triggering episodes of severe hypoglycemia, rather than achieving definitive insulin independence. To date, multiple advances and positive results at the level of glycemic control have been obtained by performing pancreatic islet transplants thanks to recent advances in improved techniques, isolation and immunosuppression methodologies [123]. These results have been positive because glycemic control has been superior to that demonstrated in therapies such as the application of subcutaneous insulin. Islets transplantation has improved substantially in last decade with multiple refinements including more optimal islets preparation, culture, safer transplant techniques and more effective anti-inflammatory and immunomodulatory interventions. Excellent therapeutic effects have been confirmed as a result of reactive insulin secretion, leading to a normalized blood glucose level, which cannot be reproduced with pharmacological insulin therapy [124,125]. However, some problems need to be resolved, such as severe donor shortages, and more research is need regarding alternative transplant sites that can be grafted efficiently, alternative cell sources, blood-mediated inflammatory reaction, and the functional preservation of β-cells [126,127].

Prior to the implementation of the DNA recombinant technique, insulin from pigs was utilized for the treatment of diabetic patients [128]. Insulin from pigs is practically identical to human insulin, with only one amino acid difference [129]. Pig pancreatic islets can be adequately applied for clinical use in humans, avoiding some of the technical limitations involved in cadaveric transplants, such as senescence, brain death, comorbidities, or cold ischemia [130,131]. Several works have been carried out in which the therapeutic effectiveness of pig islets in nonhuman primates was observed. However, technical difficulties with these xenografts, including virus transmission or interspecies antigenicity, have limited their use [132,133]. New techniques of gene editing, as well as the generation of pigs with human pancreata as the source of pancreatic islets, are under investigation [134,135]. Porcine endogenous retrovirus (PERV) is considered the major challenge of biosafety in xenotrasplantation. Editing technologies such as CRISPR-Cas9 have been used to inactivate PERV genes reducing PERV transmission to human cells. Additionally, the potential to humanize the pig genome using CRISPR-Cas9 technologies will generate more promise in xenotransplant, however increasing ethical challenges. On the other hand, microencapsulation of pig islets for later transplantation is feasible, even without previous immune therapy of the receptors, demonstrating a significant reduction for more than 600 days of hypoglycemic episodes [136,137].

The maintenance of β-cells is established by the replication capacity of the β-cells themselves. This ability progressively diminishes over time, due in part to the great functional specificity of β-cells [138,139,140]. A large number of growth factors and mitogenic agents have been shown to promote β-cell proliferation in animal models [141,142]. However, these agents have generally failed to promote significant proliferation in human β-cells. Many differences among pancreatic islets in animals relative to humans and molecular and epigenetic changes observed when β-cells reach maturation have challenged this practice [143]. Some of these changes seem to improve β-cell function, but they may broadly suppress the ability of β-cells to respond to proliferative stimuli. A potentially important advance has come from compounds such as dual specificity tyrosine-phosphorylation-regulated kinase 1A (DYRK1A) or the stimulation of β-cells by insulin or glucose [144,145].

There has been great interest in using master regulators of the development of β-cells to convert non-β-cells into insulin-producing cells [146,147]. In experimental studies in which complete ablation of β-cells is achieved, other cells of the endocrine islets of the pancreas may be differentiated to β-cells [148]. Transdifferentiation to β-cells, when the damage is severe or lasting, can occur over time. Several studies have attempted to reproduce this transdifferentiation by overexpressing specific transcription factors, such as PDX-1 and MAF bZIP transcription factor A (Mafa) [149]. Although the molecular mechanism of these conversion events remains unknown, genetic deletion of aristaless related homeobox (ARX), a regulator of α-cell development, or force expression of paired box 4 (PAX4), a regulator of β-cell development, can convert α-cells into β-cells [150,151]. Cell reprogramming from other cells, especially from the exocrine pancreas, has also shown encouraging results [152,153,154]. Transdifferentiation of acinar cells and duct cells to β-cells has been reported in mouse models [155]. Melton’s group [152] performed reprogramming of acinar cells to β-cells in vivo by polysynchronical expression of the factors Pdx1, Ngn3 and Mafa. These cells were able to form clusters similar to pancreatic islets and to secrete insulin after stimulation with different glucose concentrations [154]. This work confirmed the importance of the presence of a “niche” islet environment to effectively secrete insulin. Tissues of endodermic origin, such as those in the liver or intestines, have shown a high level of plasticity and can also be differentiated into β-cells [156,157].

## 5. Human Pluripotent Cells and DM

The use of pluripotent cells has been a research tool in frogs, fish and mice and have offered encouraging results for therapeutic approaches in regenerative medicine. These studies have successfully deciphered which regulatory pathways are highly important in different stages during the embryonic development of the pancreas [158]. Notwithstanding, in humans, it has been a technical and ethical challenge, and the results have contributed to the knowledge of the embryonic development of the pancreas [159,160]. A breakthrough in the generation of pluripotent cells from somatic cells was achieved through the overexpression of specific transcription factors [121]. Both embryonic pluripotent stem cells (ESCs) and induced pluripotent stem cells (iPSCs) are distinguished by their ability to propagate in an undifferentiated state and to differentiate into any cell type in the human body, reflecting their tremendous therapeutic potential [161]. However, the same plasticity that enables pluripotent stem cells to generate hundreds of different cell types also makes them difficult to control and presents safety considerations in terms of developing a stem cell-derived cell product [162,163]. Fortunately, developmental pathways are highly conserved among species; the signaling pathways that regulate human cell differentiation are similar to those that regulate these processes in other organisms [164]. In general, the experimental process of differentiation from a pluripotent cell has low efficiency. Many differentiated cells have dysregulated signaling pathways, and the molecular markers of each stage of the differentiation process have no yet been identified [165]. The development of protocols for the differentiation of pluripotent cells towards insulin-producing β-cells in vitro requires a series of molecular signals that can be detected in vivo models [166,167] (Figure 3).

In protocols for the development of β-cells from pluripotent cells, six stages have been established using specific inducing factors to produce islets containing β-cells. Specific markers determined by flow cytometry or immunofluorescence microscopy are used to assess the progress and efficiency of the differentiation process [168,169,170]. The first three stages of differentiation generate a nearly homogenous (approximately 90% clonal) population of progenitors that express the master transcription factor PDX1. Thereafter, distinct populations are identified by staining for C-peptide, the pan-endocrine marker chromogranin (CHGA) and the β-cell transcription factor NKX6.1 [91]. Endoderm induction from pluripotent stem cells has been studied in animal models, and among the most important factors that regulate this initial step towards the germ line is the TGF-β superfamily of growth factors [171]. Generation of the final endoderm is determined by the expression of forkhead box protein A2 (FOXA2) and SRY-box transcription factor 17 (SOX17). Following the appearance of the definitive endoderm, the pancreatic endoderm is determined by the presence of the PDX1, NKX6.1, SOX9, and pancreas-associated transcription factor 1a (PTF1A) factors [172]. The cells of the pancreatic endoderm then divide into the endocrine pancreas or exocrine pancreas. The ducts of the exocrine pancreas maintain the expression of SOX9, and the acinar cells maintain the expression of PTF1A [173]. These cells will be located along the structures that resemble buds, while endocrine cells will form in the trunk of these buds. The endocrine progenitor cells maintain the expression of PDX1 and NKX6.1, but above all, they induce the expression of NGN3 and NEUROD1. Between 2005 and 2008 Baetge’s group [171,174,175], developed techniques to differentiate hESCs towards β-cells. In 2005 they developed a low-serum medium supplemented with Activin A in monolayer cultures with feeder cells for 5 days. They differentiated these cells towards the final endoderm. Years later, they extended their protocol for the generation of insulin-secreting endocrine cells that respond to various glucose concentration [174]. The use of retinoic acid (RA), as well as the inhibition of sonic hedgehog via cyclospamine and the addition of FGF10, allowed them to obtain cells that expressed PDX1, NGN3 and insulin and showed a good response to glucose [176]. Incorporating sodium butyrate which increases the action of Activin A [177] and modulates the WNT, BMP and TGF-β pathway [178], improved the yield of hβ-cells derived from stem cells. Modification of the protocols to specifically produce endocrine cells by culturing the cells with epidermal growth factor (EGF), Noggin, and nicotinamide formed a population of 70% cells expressing NKX6.1 [179]. The introduction of vitamin C improved the production of cells that express MAFA and increased the efficiency of differentiation due to epigenetic modification [169,180]. Vitamin C is known to have the ability to modulate histone demethylases and may improve the expression of some genes [181,182]. They also introduced triiodothyronine (T3) and a transforming growth factor b receptor 1 inhibitor (ALK5/TGF-bR1). With this protocol, β-cells were produced that responded to glucose but in a delayed manner [183]. Melton’s group introduced a sonic hedgehog pathway antagonist (SANT1), and after obtaining progenitor pancreatic cells, they introduced LDN, (a BMP type 1 receptor inhibitor) T3, XXI (a γ-secretase inhibitor), heparin, alk5 receptor II inhibitor (alk5i), and β-cellulin (member of the EGF family) [91,168]. Recent identification of cells obtained after the differentiation of pluripotent stem cells showed cellular heterogeneity, with specific groups of cells in different stages. Progenitors of pancreatic cells were identified in stage 3 and stage 4. At the end of stage 4, NKX6-1 progenitors, as well as the first α-type cells, were observed. Finally, in stage 5 and 6, three classes of CHGA endocrine cells were observed: (a) β-cells expressing Ins, Nkx6-1, Isl1, and other β-cell markers; (b) α-cells expressing glucagon (Gcg), Arx, Iroquois homeobox 2 (Irx2) and Ins; and (c) a type of endocrine cell that expresses Chga, tryptophan hydroxylase 1 (Tph1), LIM homeobox transcription tactor 1 Alpha (Lmx1α), and C-type lectin domain family 18 (Lc18a1), which most closely resembles entero-chromaffin cells [91,184,185]. The observation of enterochromaffin-like cells within the islets, which showed similarities to β-cells, suggests that there is a relationship between the fates of β-cells and enterochromaffin cells. This is a step forward in recognizing the complex web of cell physiology within the pancreatic islets. The description of differentiation stages of pluripotent cells has made it possible to specify the maturation process of β-cells in vivo. However, the function of the transplanted cells requires a greater recognition of the events that lead to the close coordination between insulin secretion and serum glucose levels. In general, mature β-cells secrete similar levels of insulin in response to glucose and potassium. While, immature β-cells do not have a coordination between these events. It was recently shown that endocrine cell clustering or expression of mitochondrial activity regulators induces metabolic maturation by driving mitochondrial oxidative respiration, a process central to insulin secretion in mature β-cells. Giving hope towards obtaining mature β-cells [185,186,187].

Multiple preclinical studies in β-cell generation have been developed to date and, most of them have been focused on type 1 DM where the β-cells obtained have been derived from both healthy and diabetic individuals [188,189,190]. Millman et al. [191], demonstrated that there was no difference between β-cells obtained from diabetic and non-diabetic patients in order to transplant β-cells into type 1 diabetic patients. Additionally, an increasing number of subjects with type 2 DM have a complete deterioration of β-cells function and require insulin therapy. Some new studies have shown that the effect of stem cell therapy may improve β-cell function in type 2 DM [192,193,194]. However currently, stem cell therapy is indicated for young type 1 DM subjects complicated by impaired awareness of hypoglycemia and recurrent severe hypoglycemia, extreme glycemic lability. In general, the diabetic patients with “brittle” diabetes [185]. Some of the criteria that have been taken into account to exclude patients by the initial clinical trials are a-) Body mass index (BMI) > 33; b-) Insulin requirements > 1.2 units/kg/day; c-) Significant kidney dysfunction. However, patients who have already had a kidney transplant may be candidates for the therapeutic strategy of pluripotent stem cells. Other clinical circumstances that exclude the therapy with Pluripotent Stem Cells are: high blood pressure despite appropriate treatment, high cholesterol/triglycerides despite appropriate treatment, anemia or other blood disorders that require medical treatment, increased risk of bleeding, other chronic hemostasis disorders, or treatment with chronic anticoagulant therapy. Psychiatric illness that is untreated, or likely to interfere significantly with study compliance despite treatment [195,196].

One of the most striking challenges of β-cell transplantation is preserving functionality of cells after the transplant [197]. Two main strategies have been evaluated, encapsulation of the cells and the use of scaffolds with integrative action with the host cells. The encapsulation of pluripotent stem cells allows oxygen, nutrients, proteins, and other molecules necessary for cell survival and function to reach the cells and allows insulin and other proteins secreted by the cells to exit the device. Encapsulation devices must prevent damage caused by the autoimmune process in patients with type 1 DM and permit the differentiation and maturation of β-cells. Various materials have been studied to produce the devices, and many of them are composed of alginate [198,199,200]. In contrast to encapsulation strategies, recent approaches aim to use scaffolds that allow the transplant to be integrated with the host to facilitate glucose detection and insulin distribution. Scaffolds can provide a supportive niche for cells during the manufacturing process or after transplantation at extrahepatic sites. Scaffolds require therapy to evade the host’s immune response, while having the ability to deliver oxygen, angiogenic or trophic factors, and can facilitate co-transplantation of cells, which enhance engraftment or modulate immune responses [201,202] (Figure 3).

IPSCs can be reprogrammed from somatic cells of the patient and differentiated for application in the affected tissue [203]. The use of this type of cell has the advantage of ostensibly reducing the probability of immune rejection in the recipient and, similarly, avoiding the ethical problems related to the use of embryonic pluripotent cells [204,205]. Treatment with iPSCs is based on the properties of specific proteins of pluripotent cells, which when overexpressed can, reprogram somatic cells. This task can be achieved with the transcription factors, octamer-binding transcription factor 4 (Oct4), Krüppel-like factor 4 (Klf4), Sex determining region Y-box2 (Sox2) and oncogene of myelomatosis (c-Myc) (OKSM), which have been recognized for preserving the pluripotency of a staminal cell [206,207]. These factors support the capacity of an already differentiated cell to shift to a pluripotent state. Moreover, the factors associated with the specific processes of different germ lines and specific cell groups can be exploited to improve the function of damaged tissue [206]. Several studies on iPSCs have revealed that they can generate the angiogenic process completely, allowing the restoration of normal blood flow to the affected tissues where they have been tested, indicating that they are a fundamental and alternative source of vascular tissue even for the treatment of wounds in the human body [208].

The generation of iPSCs is carried out with methods based on viral vectors and nonviral vectors. The reprogramming efficiency is variable depending on the method used. In general, virus-based methods lead to a high efficiency, genome integration but do have safety limitations [209]. Most iPSCs are made using retrovirus vectors, which integrate reprogramming factors. Retrovirus vectors can spontaneously infect cells and insert their reprogramming genes into the host’s genomes using reverse transcriptase, which allows transgenic expression to continue. Retroviral transgene expression continues until the cells become iPSCs, and then the retroviral promoter is inactivated, possibly due to epigenetic modifications such as the methylation of histones [210]. These guided reprogramming and automatic silencing mechanisms are considered very important for the induction of iPSCs from somatic cells. Several groups have tried to establish reprogramming methods that do not require viral vectors trying not to affect the efficiency of the process [211,212,213]. On the other hand, compared to pluripotent embryonic cells, iPSCs have some differences in gene expression. Epigenetic modifications of iPSCs lead these cells to maintain a “memory” of the somatic cell from which it originated. This circumstance could then affect the differentiation towards the cells that are to be replaced [214,215,216].

Previous observations have shown caution in studies with iPSC cells due to the possible presence of immunological rejection even in autologous transplants. The abnormal expression of several genes of the iPSCs showed the induction of an immune response mediated by T cells. Despite the results obtained by Zhao et al., [217], most of the reports to date have shown no or lack immunogenicity in the use of autologous iPSCs in animal transplants. Beyond the problems that may presented during iPSCs transplants, it is important to ensure the establishment of standard protocols for reprogramming processes, since most of the problems in the functionality or increased tumorigenic of iPSCs are due to secondary alterations of the cell genome [218]. Although currently reprogramming technology allows the generation of autologous iPSCs for specific patient treatments in a relatively affordable way. The process to obtain autologous iPSCs is laborious, expensive, associated with the possibility that the quality and efficacy of the cellular products of each individual is uncertain. Therefore, most regenerative approaches that rely on the generation of autologous iPSCs have been abandoned [219,220]. Allogeneic iPSC cell therapies targeting large patient populations may be more economically feasible but are subject to strong immune rejection. For allogeneic transplantation, although iPSC pools isolated from homozygous HLA donors are being considered to cover the majority of HLA haplotypes, the recruitment of homozygous HLA donors serving an entire population is not trivial [221]. The cell bank with homozygous HLA-A, -B, -DR iPSC lines to have a coverage of more than 80% of the population can be very large, especially in multiethnic societies [222]. Considering life-time immune suppression in allogenic pluripotent stem cell-based transplant, the need for a universal solution to immune incompatibility is a regenerative medicine priority. The gene editing of the HLA system molecules and other molecules that improve the compatibility of the immune response may be a strategy to prevent the rejection of allogenic transplants. However, it is a research field that is in its beginning, since a reduction in the antigenic capacity of cells can increase tumorigenicity and predisposition to infections [223,224].

Taking into consideration the findings of different researchers, some limitations regarding use of pluripotent stem cells remain: (a) Stem cell-derived beta cells do not have robust physiologic function; (b) More efficient low-cost differentiation methods are needed; (c) the elements that control the differentiation of the different cells inside of the pancreatic islets and their proportions must be further clarified; (d) the techniques used to perform the transplant must be improved; (e) Devices that maintain cells and allow them to establish a physiological function preventing immune attack must be standardized; (f) Likewise, the implantation site of the devices must be established; (g) Because pluripotent cells have the property of giving rise to cells of the different germ lines, a latent risk of the transplantation of these cells is the development of teratomas [217,225]. New gene editing techniques or related methods that can ensure immunological evasion of the transplanted cells, increase the functionality and viability of the cells must be incorporated and finally the metabolic, energetic and nutritional environment of β-cells produced from pluripotent cells must be controlled. One of the main drawbacks with the present techniques for the production of iPSCs and their subsequent differentiation in specific tissues is that the populations of interest are small in size [185,226]. It is probable that this low yield of induced pluripotent cells is a consequence of the development of an epigenetic barrier that limits the differentiation process [227,228,229,230].

## 6. Epigenetics and Mechanisms of Chromatin Modification

The differentiation of ESCs requires a series of chromatin rearrangements to establish cellular identity. The scope is even more complex when inducing differentiation from a somatic cell to a pluripotent stem cell to the desired cell type by manipulating specific genes [231,232,233]. Heterochromatin has two functional categories: the constitutive heterochromatin, which is located in strongly compacted areas formed in many cells in the centromeres and in the telomeres, and facultative chromatin, which has location and cellular specificity [234,235]. Constitutive heterochromatin stops the recombination of conserved genomic areas on chromosomes. Facultative heterochromatin silences genes that encode proteins that are not part of the cell’s functional identity [236]. Current models of embryonic cell differentiation suggest that compacted heterochromatic domains expand as cells differentiate, helping to establish cellular identity [237]. Posttranslational modifications of histones usually regulate the dynamics of heterochromatin. Histones are subject to various posttranslational modifications, such as acetylation, methylation, phosphorylation, and ubiquitination, and thus contribute to the regulation of chromatin states and transcriptional activities [238]. The chemically stable characteristics of methylated histones contribute to the cellular memory of external stimuli by maintaining the transcription levels of adaptive genes even after the dissipation of environmental cues [239]. The most important repressive transcriptional modifications are the methylation of histone 3 at lysine 9 and lysine 27 residues (H3K9me and H3K27me) [234,240]. In mammals, euchromatic histone-lysine N-methyltransferase 2 (EHMT2/G9a) and G9a-related methyltransferase (GLP/Eu-HMTase1) predominantly regulate H3K9 mono- and dimethylation and are essential for embryogenesis [241,242,243]. SET domain bifurcated histone- lysine methyltransferase (SETDB1), lysine methyltransferase 1A (KMT1A/SUV39h1) and lysine methyltransferase 1B (KMT1B/SUV39H2) catalyze H3K9 trimethylation (H3K9me3) [244,245]. However, the tissue specificity and regulatory genes related to these factors are not yet clear; in vivo studies have yielded possible redundant functions. H3K9me2/3 are recognized by heterochromatin protein 1 (HP1), which, through auto-oligomerization and interaction with other repressive modifications, ensures the compaction, spread and inheritance of heterochromatin [246,247,248]. The repressive platform established by the H3K9me2/3-histone and HP1 methyltransferases favors the establishment of DNA methylation and the maintenance of low acetylation in histones [249,250]. Regions of trimethylated histone 3 lysine 9 (H3K9me3)-marked heterochromatin can have a physically condensed structure that serves to silence protein-coding genes at facultative heterochromatin [234]. In particular, specific genes of the differentiated cells derived from the endoderm showed a net loss of H3K9me3 throughout the hepatic and pancreatic lineages [250]. Polycomb repressive complex 2 (PRC2) deposits methyl groups on H3K27, which can block transcription initiation. PRC2 has been shown to be important for the functional regulation of β-cells in the pancreas and for the dedifferentiation of α-cells and β-cells. It is possible that the metabolic alterations that induce the appearance of type 2 DM have an influence on the regulation carried out by PRC2 in the chromatin structure [162,251]. The activity of H3K9 and H3K27-HMTs is counterbalanced by erasers from the jumonji (JmjC) domain-containing demethylase families, with the JMJD2/KDM4 family displaying activity towards H3K9me2/me3 residues (as well as methylated H3K36), and the JMJD1/KDM3 proteins displaying activity towards H3K9me2/1 [252,253,254,255,256,257]. The KDM6A/UTX and KDM6B/JMJD3 proteins can function through their catalytic domain as histone demethylases at H3K27 residues. Both proteins have shown an important role in cellular differentiation [258,259] (Figure 4).

Other modifications of chromatin may play an important role in the complete maturation of pancreatic islet cells and, according to functional reciprocity, establish a pattern of secretion that simulates the regulation of insulin secretion in healthy subjects. MicroRNAs (miRNAs), a class of endogenous small noncoding RNAs in eukaryotes, have been recognized as significant regulators of gene expression through posttranscriptional mechanisms. MiRNAs regulate insulin secretion, pancreatic development and β-cell differentiation. However, the function of miRNAs in the formation of insulin-producing cells from adult stem cells is poorly understood [260,261].

DNA methylation plays a critical role in β-cell development and gene regulation. Several reports have highlighted the pivotal role played by key epigenetic modifiers, such as DNA methyl transferase 1(Dnmt1) during pancreatic development and how their dysregulation may lead to diabetes progression. For instance, β-cell specific Dnmt3A-konck-out (KO) mice show aberrant expression of major developmental metabolic genes, such as hexokinase 1 (hk1) and lactate dehydrogenase A (ldhA), resulting in defective glucose-stimulated insulin secretion in the postnatal stage [262]. The DNA methylation state of the promoters of genes regulating pancreatic specification such as INS, is also important for regulating β-cell mass and functions during aging [263]. Inactivation of the β-cells-specific DNA methyltransferase Dnmt1 leads to loss of β-cell mass and concomitant trans-differentiation into α-cells [264].

## 7. Conclusions

The progressive increase in DM has led to a continuous search for new therapeutic approaches to reduce the chronic complications of this disease. Nearly a century after the discovery of insulin, most type 1 DM patients and type 2 DM patients who require insulin still cannot achieve proper glucose level control. This circumstance has dire consequences, as complications begin to appear in a relatively short period of time. The management of diabetes and its complications, both micro- and macrovascular, has a great impact on individuals and families as well as public health systems. Although external devices for the monitoring and administration of insulin have improved glycemic control and metabolic control in patients with diabetes, insulin is so tightly regulated by the body that it still exceeds the capabilities of increasingly sophisticated forms of subcutaneous administration.

Regenerative medicine therapy using pancreatic islet cells can be expected to achieve an adequate response to changes in blood glucose levels and immediate secretion of not only insulin but also multiple hormones necessary to maintain metabolic control. In the last decade, iPSCs have opened an extraordinary new avenue for therapeutic strategies for the patients with diabetes. However, time is a factor working against molecular biology for the further development of these strategies; issues related to various immunological factors, epigenetic memory, and the maturation of insulin-secreting cells that limit the establishment of iPSCs as a therapeutic method must be resolved.

Despite these difficulties, a study of a cohort of 200 patients with type 1 diabetes undergoing treatment with the product VC-01, which is derived from pluripotent stem cells, is ongoing. VC-01 consists of pancreatic endoderm cells derived from human ESCs, encapsulated in ViaCyte’s Encaptra device (ClinicalTrials.gov identifier: NCT02939118) [265]. Since DM does not currently have a specific cure, treatment with regenerative medicine via the use of induced pluripotent cells (iPSCs) has emerged as possible ideal treatment for DM patients [231,266,267].

## Figures and Tables

**Figure 1 ijms-21-08685-f001:**
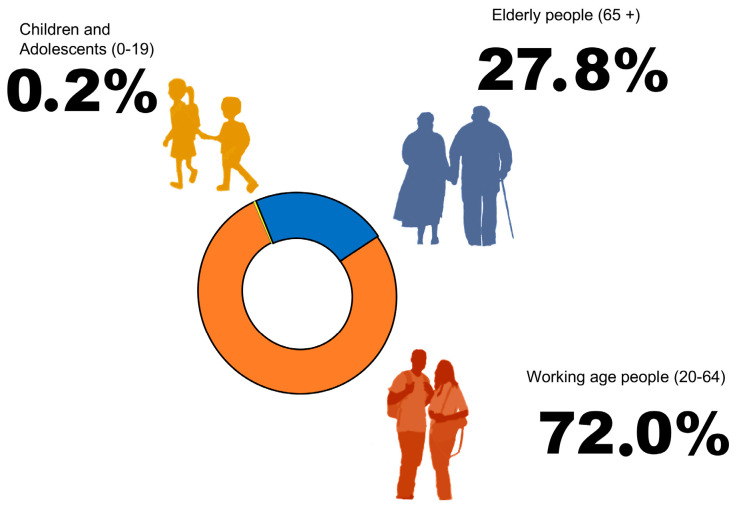
Diabetes mellitus is a global problem. A. The estimated number of people aged over 65 y with DM was 111 million in 2019. (1:5 people of this age are predicted to have DM). It is projected that by 2045, the number of people aged over 65 y with DM will reach 276 million. B. An estimated 1.1 million children and adolescents (aged under 20 y) have type 1 DM. Overweight and obesity are increasing the number of children and adolescents with type 2 DM. C. Three in four people living with DM are working age (aged 20–64 y), and this number is expected to increase to 486 million by 2045. IDF, Atlas of Diabetes 2019 [49].

**Figure 2 ijms-21-08685-f002:**
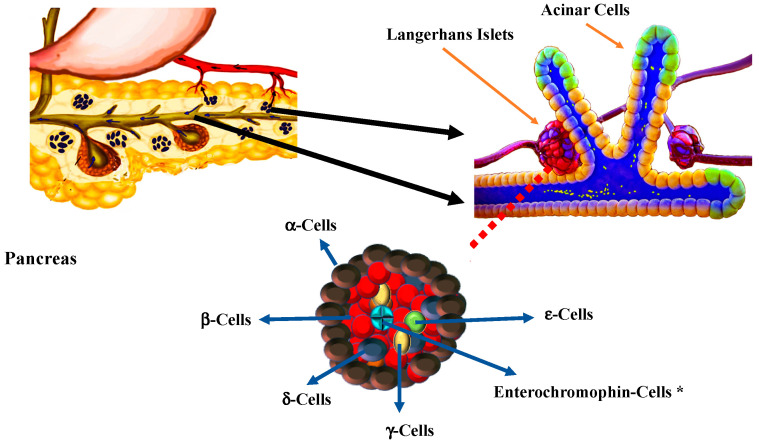
The pancreas has two main functions. As an exocrine organ, it secretes enzymes to digest proteins, carbohydrates and fats. The endocrine component of the pancreas consists of islets of Langerhans that create and release hormones directly into the bloodstream. Pancreatic endocrine cells are clustered into islets of Langerhans, which are composed of different functional cells. The endocrine pancreas consists of 5 main types of secretory cells, alpha cells (α-cells), which express glucagon; beta cells (β-cells), which express insulin; delta cells (δ-cells), which express somatostatin; gamma cells (γ-cells), which express pancreatic polypeptide; and epsilon cells (ε-cells), which express ghrelin. In adults, 60% of islet cells are β-cells, and 30% are α-cells. Different types of endocrine cells establish a paracrine network and interact with peptides involved in an autocrine-mediated hormone secretion. * Enterochormaphin cells have been described in human in vitro β-cells differentiation [91].

**Figure 3 ijms-21-08685-f003:**
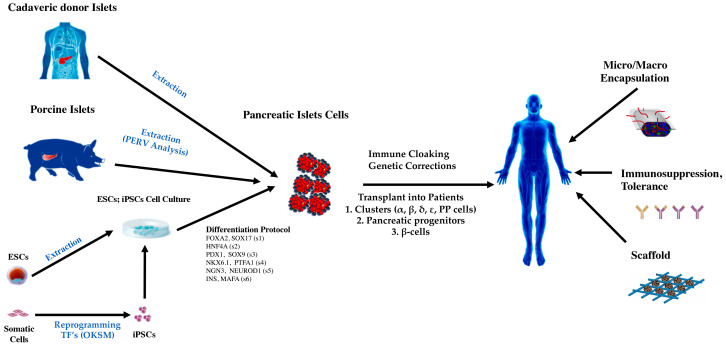
Islets obtained from the human or porcine pancreas can be transplanted to subjects with diabetes, following some form of immunosuppression. Human pluripotent stem cells can be differentiated into islet progenitor cells, pure beta cells, or mixed islet cells prior to implantation. Both donor pigs and pluripotent stem cells can be genetically engineered to improve survival, function, and cell safety. Cells can be micro, or macro encapsulated prior to implantation to reduce the requirement for chronic immunosuppression. The cells can also be incorporated into a scaffold to support cell survival and graft recovery. Differentiation Protocol is divided in sequential stages (s): FOXA2, Forkhead boxA2 and SOX17, SRY-box transcription factor 17, stage 1 (s1); HNF4a, Hepatocyte nuclear factor 4 alpha, stage 2 (s2); PDX1, pancreatic and duodenal homeobox 1 and SOX9, SRY-box transcription factor 9, stage 3 (s3); NKX6.1, Nirenberg and Kim homeo box 6.1 and PTFA1, pancreas associated transcription factor 1a, stage 4 (s4); NGN3, Neurogenin 3 and NEUROD1 Neuronal differentiation 1, stage 5 (s5); INS, insulin and MAFA, V-maf musculoaponeurotic fibrosarcoma oncogene homolog A, stage 6 (s6); ESC, embryonic stem cells; iPSC, induced pluripotent stem cells; TF’s: transcription factors; OKSM: octamer-binding transcription factor 4 (Oct4), Krüppel-like factor 4 (Klf4), Sex determining region Y-box2 (Sox2) and oncogene of myelomatosis (c-Myc).

**Figure 4 ijms-21-08685-f004:**
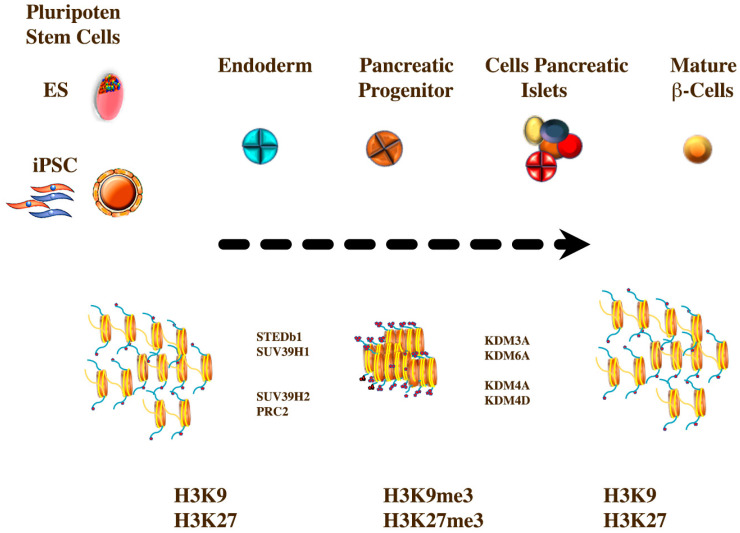
Graphic representation of the influence of the methylation pattern on pancreas development. The development of pancreatic islet cells are subject to a number of epigenetic variations. The modifications of the methylation of histones 3 in lysine residues 9 and 27 (H3K9 and H3K27) stand out. The highest number of genes marked by H3K9me3 is detected at the germ layer stage in endoderm. SET domain bifurcated histone-lysine methyltransferase (STEDb1), Suppressor of Variegation 3-9 Homolog 1, -2 (SUV39H1), -SUV39H2 and the Polycomp complex (PRC2) produce hypermethylation of the lysine residues (H3K9me3 and H3K9me3). After the differentiation of the Endocrine pancreas cells. Progressive loss of H3K9me3 and H3K27me3 at lineage-specific genes mediated by lysine demethylases, KDM4A, -KDM4D, -KDM3A, -KDM6A.

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
