# Peer review of "Diabetes Mellitus Is a Chronic Disease that Can Benefit from Therapy with Induced Pluripotent Stem Cells"

_ijms, 2020, doi:10.3390/ijms21228685_

Round 1

Reviewer 1 Report

In the review paper entitled “Diabetes mellitus is a chronic disease that can benefit from therapy with induced pluripotent stem cells” Arroyane and co-workers focused their attention on potential benefits of stem cells treatment for diabetes providing a detailed report on induced pluripotent stem cells (iPSCs) generation and differentiation. Moreover Authors described the iPSCs isolation protocol with particular regard to molecular inducible markers expression to ensure a stable and correct differentiation towards beta cell like phenotype. Since regenerative medicine represents the new approach in metabolic disease management, this review can be useful for scientists whose work implies the use of those kind of cells.

The paper is well written, figures are clear and figure legends are exhaustive. References are recent and published on prestigious journals.

Nevertheless, some parts of this review deserve a more specific explanation:

  • in paragraph 4 row 249-257 Authors reports about the failure in performing islet transplantation: this particular problem needs to be deeply explained since lot of groups tried several strategy to solve the problems related to transplantation. Discussing these possible solutions and the approach they used could be will be very interesting.
  • Several recent works have highlighted the potential iPSCs  immuno toxicity since it has been demonstrated that even autologous stem cells are potentially capable to unleash immune host reaction. These data deserve at least a paragraph in this review.

Author Response

  1. In paragraph 4 row 249-257 Authors reports about the failure in performing islet transplantation: this particular problem needs to be deeply explained since lot of groups tried several strategies to solve the problems related to transplantation. Discussing these possible solutions and the approach they used could be will be very interesting.

Thank you very much for your comment. You are right the transplant of b-cell it is not a failure procedure.  The problem with b-cell is that the number of cells obtained from a donor are limited. Thus, to be successfully it is necessary a lot of donors. We performed the correction in the text of the manuscript.

  1. Several recent works have highlighted the potential iPSCs immuno toxicity since it has been demonstrated that even autologous stem cells are capable to unleash immune host reaction. These data deserve at least a paragraph in this review.

Thank you very much for your comments. It is an interesting point, following your recommendation a new version is in the manuscript.

Reviewer 2 Report

General comments:

Arroyave and colleagues have written a general purpose review that broadly covers some aspects of regenerative medicine with regard to pluripotent stem cells. General purpose because the authors have put their analysis on a very broad fundament. Next to the obligatory introduction section to the disease of diabetes mellitus the reader can learn a lot e.g. how diabetes burdens the public health systems, the genetics of diabetes, pancreatic organogenesis, current prospects of regenerative medicine, iPSC & reprogramming, development of differentiation protocols for PSCs and epigenetics and differentiation. Thereby the authors contribute very little to the goals of their own review, namely to inform the reader, how diabetes patients can benefit from a pluriopotent stem cell therapy. In many ways the authors only scratched on the surface of the topic and miss to discuss the major issues and obstacles of this particular field right now. Important questions are only briefly discussed or left out. E.g. which patients are the target for a stem cell therapy and which are not Why is the cell implant being encapsulated in the current clinical study by Viacyte? What do we know about the current limitations of the stem cell therapy? Any mentionable safety aspects? How is the transplant material composed? How will the persisting autoimmunity in T1DM effect the therapy goals? I could add more point but I hope the authors get the idea of the reviewer’s criticism. The review is not focused enough. On top of that the current state of the manuscript precludes publication.

Some major and minor problems of the manuscript without any meaningful order:

  1. Beta cells, b-cells or β-cells. The authors should stick with on designation and then keep it (preferably beta cells).
  2. Line 43. Impaired is the wrong wording, destroyed would fit better tot he mechanisms of T1DM
  3. Line 54-57. Nature does only now normal insulin. Short- and long-acting insulins have been engineered to simulate phyiological effects.
  4. Line 57. Remove every day as this is factually wrong and colloquial.
  5. Line 79. Reprogramming is an universal principle and does not require a cellular niche. Unless this is not further explained I would recommend to remove this sentence.
  6. Line 84-159, this para should be carefully revised with respect to the different acute and chronic complications of T1DM and T2DM.
  7. Figure 1. Remove the commas with periods.
  8. In Figure 3. I don't see any "series of molecular signals"
  9. Line 160-172. This para refers to the polygenetic nature of T2DM and mentions some risk-associated gene loci for T1DM. This specific section is too superficial to be useful for the reader. A short sentence mentioning polygenetics and T2DM should be enough.
  10. Line 249-267, whole pancreas transplantation and islet transplantation have different therapy goals. The authors should take that into account. Pig islets are worth mentioning but the link between xenotransplantation and genetic editing is not clear. Please extend.
  11. In general, everything is a bit mixed up when talking about the development of beta cells, it is more logical to continue with in vitro differentiation, instead it is interrupted by another topic.
  12. Maybe ESCs and iPSCs should also be compared more closely. ESCs are only mentioned briefly. Are iPSCs more useful than ESCs when developing a therapy for autoimmune diseases?
  13. References: This reviewer checked several references and they were wrong. It is recommended to check all citations and the bibliography of the manuscript. Example refs #147, 148, 149 and 114 refer to the wrong publications.
  14. Line 311/312. This is not correct. Efficiency is generally not bad. Cardiomyocyte and neuronal differentiation is highly effective. The general problem of generated cells is maturity. Maturity and polyhormonality is another important aspect that is not covered in the review.
  15. Line 326-328. Please name the surface markers.
  16. Line 374-377. The question of immungenecity of differentiated iPSC is a very sensible one. Data have been published reporting both: complete immunogenetic identity and also partial rejection of differentiated iPSC in syngenic transplantation models. Please revise this para.
  17. Conclusions: As mentioned above. Reading the conclusion, it is not quite clear how a diabetic patient can take advantage of a PSC-based therapy today. The Viacyte clinical study started in 2014 and was projected for 3 years. A final report has not been published. There is a reason for the delay.
  18. The last sentence is factually wrong. Till the writing of this peer review not a single study supports the assessment that iPSC or ESC based therapies have emerged as an ideal treatment for DM patients.

Author Response

  1. Beta cells, b-cells or β-cells. The authors should stick with on designation and then keep it (preferably beta cells).

Thank you very much for your observation. The words were designated singly according to your recommendation. All words were homologated to b-cells.

  1. Line 43. Impaired is the wrong wording, destroyed would fit better to the mechanisms of T1DM

Thank you. It is really what happened in Type 1 DM, b-cells are destroyed for the immune attack. We change the word.

  1. Line 54-57. Nature does only now normal insulin. Short- and long-acting insulins have been engineered to simulate phyiological effects.

Thank you for your observation. We believe that it was misunderstood all the sentence. The insulin that is produced by the pancreas is just one insulin. The intended expression is that by genetic engineering and recombinant technics it is possible to achieve different kinds of insulin. All para is to highlight the importance of recombinant insulin for humankind. 

“At present, some amino acids in this recombinant insulin have been modified to reestablish the physiological effect of endogenous insulin using short-, intermediate- or long-acting insulins”

  1. Line 57. Remove every day as this is factually wrong and colloquial.

Thank you. The word was removed.

  1. Line 79. Reprogramming is a universal principle and does not require a cellular niche. Unless this is not further explained I would recommend removing this sentence.

Thank you again. The sentence is no referred to as induced pluripotent cells (hiPSCs), is about human pluripotent cells (hPSCs). Some tissues in adults have some cells with pluripotency, in order to renew functional cells for the normal function of the tissue. Curiously tissues such as the pancreas or the intestine do not have this type of pluripotent cells. In the pancreas, renewal can be carried out from other cells of the islets or from acinar cells.

  1. Line 84-159, this para should be carefully revised with respect to the different acute and chronic complications of T1DM and T2DM.

Thank you for your observation. Acute and Chronic complications are the main problem o diabetic patient. We intended just a brief note about the complications in order to show the importance to look for a cure for the disease. The paragraph has been carefully reviewed and the corresponding clarifications have been made.

  1. Figure 1. Remove the commas with periods.

Commas have been removed at your suggestion

  1. In Figure 3. I don't see any "series of molecular signals"

Thank you. We performed corrections

  1. Line 160-172. This para refers to the polygenetic nature of T2DM and mentions some risk-associated gene loci for T1DM. This specific section is too superficial to be useful for the reader. A short sentence mentioning polygenetics and T2DM should be enough.

Thank you very much for your observation. The paragraph between lines 160-162 has been modified by a short sentence. Since we are interested in leaving a broad vision about diabetes, we would like to include a brief review of the susceptibility to diabetes mellitus.

  1. Line 249-267, whole pancreas transplantation, and islet transplantation have different therapy goals. The authors should take that into account. Pig islets are worth mentioning but the link between xenotransplantation and genetic editing is not clear. Please extend.

Thanks again. A porcine endogenous retrovirus (PERV) is considered the major challenge of biosafety in xenotrasplantation. Editing technologies such as CRISPR-Cas9 have been used to inactivate PERV genes reducing PERV transmission to human cells. Additionally, the potential to humanize the pig genome using CRISPR-Cas9 technologies will generate more promise in xenotransplant, however increasing ethical challenges. On the other hand, microencapsulation of pig islets for later transplantation is feasible, even without previous immune therapy of the receptors, demonstrating a significant reduction for more than 600 days of hypoglycemic episodes.

  1. In general, everything is a bit mixed up when talking about the development of beta cells, it is more logical to continue with in vitro differentiation, instead it is interrupted by another topic.

 The text was reorganized according to your suggestion to avoid the observed interruptions.

  1. Maybe ESCs and iPSCs should also be compared more closely. ESCs are only mentioned briefly. Are iPSCs more useful than ESCs when developing a therapy for autoimmune diseases?

Thank you for your suggestion. Both iPSCs and ESCs have been researched for the treatment of autoimmune diseases. iPSCs obtained through reprogramming processes are in fact like ESCs since when they grow in the culture they are characterized by their rounded shape, long nucleus, and scant cytoplasm. iPSCs offer three potential pathways through which the treatment of autoimmune diseases can be performed. The first one is the ability to provide a large number of pure cells lost during the disease (immuno-reconstitution), the second one by immune modulation of the disease in vivo, and the third one by modulating the effects of the disease in vitro. On the other hand, ESCs have been used in several studies due to their various immunomodulatory functions. However, ethical complications of using this type of cell have blocked the progress of this technique. Currently, mesenchymal stem cells (MSCs) are the most widely used and researched in this field. Studies conducted between 2003-2005 showed that MSCs can interact with most cells of the innate immune system through direct interactions between cells or indirectly through the secretion of immunomodulatory factors that enhance the function of affected cells. Despite the great advantages that these types of cells have presented in the treatment of autoimmune diseases, the major complication in their use is the risk of teratoma formation due to problems in the differentiation process.

  1. References: This reviewer checked several references and they were wrong. It is recommended to check all citations and the bibliography of the manuscript. Example refs #147, 148, 149 and 114 refer to the wrong publications.

The references were corrected according to your suggestion

  1. Line 311/312. This is not correct. Efficiency is generally not bad. Cardiomyocyte and neuronal differentiation are highly effective. The general problem of generated cells is maturity. Maturity and polyhormonality is another important aspect that is not covered in the review.

The maturation processes are a crucial stage in the studies of cells differentiated from iPSCs since, once they begin to express the characteristic markers of a specific lineage in vitro, they could be ready to be grafted and corroborate their behavior in vitro. However, to date, no conclusive report has been found that shows the different maturation processes that a cell should have after the in vitro differentiation process. To date, the only reports associated with this crucial maturation step are only hypotheses and are based on the use of co-factors or growth factors in the culture media used in in vitro trials to achieve the required viability. Pagliuca et al (2014) assert that correct modifications and proportions of cofactors and growth factors to the culture media are key to generating differentiated cells that resemble functional cells. On the other hand, they claim that many times the differences in gene expression found in research studies may be due to the death of part of the cells in the in vitro differentiation processes, the isolation and recovery processes and finally by the effect of other factors not yet determined that prevent the complete maturation of the differentiated cells. Finally, the processes of vascularization after the grafting of cells in vivo studies have a crucial role in the maintenance and viability, and this is perhaps one of the greatest challenges to be overcome when considering this type of therapies for the treatment of diseases. Thus, the omentum can be considered an ideal site for the grafting of beta cells given its high vascularization and minimal invasion during surgical processes.

  1. Line 326-328. Please name the surface markers.

The surface markers were added to the text according to their suggestions. Most of the specific markers are mentioned throughout the manuscript in the different states of differentiation of beta cells.

  1. Line 374-377. The question of immungenecity of differentiated iPSC is a very sensible one. Data have been published reporting both: complete immunogenetic identity and also partial rejection of differentiated iPSC in syngenic transplantation models. Please revise this para.

The paragraph has been revised and data from studies where the immunological rejection of iPSC transplants has occurred have been added.

  1. Conclusions: As mentioned above. Reading the conclusion, it is not quite clear how a diabetic patient can take advantage of a PSC-based therapy today. The Viacyte clinical study started in 2014 and was projected for 3 years. A final report has not been published. There is a reason for the delay.

Thank you for your comment. Regarding your inquiry about the study conducted by ViaCyte, we can inform you that after a review, the study was suspended in the year 2019 due to the body's immune reaction to the foreign body graft. Given the evidence found, ViaCyte decided to make a collaborative research partnership with W. L. Gore to investigate new materials and designs of the grafts and it is expected that between January and November of 2021 the final results of the study can be obtained.

  1. The last sentence is factually wrong. Till the writing of this peer review not a single study supports the assessment that iPSC or ESC based therapies have emerged as an ideal treatment for DM patients.

Thanks for your appreciation. We wrote that iPSC is “ideal”, actually it is no meaning that this is practical or is secure for the current treatment of DM.

Round 2

Reviewer 2 Report

Important points of criticism from this reviewer were not taken into account in the revised version of the review.

  1. Which type of patient (type 1, 2, age, gender, medical history, medical condition) is the target for a pluripotent stem cell therapy and who no?
  2. Why is the cell implant being encapsulated in the current clinical study by Viacyte? One reason is to reduce the requirement for chronic immunosuppression but their is another one.
  3. What do we know about the current limitations of the pluripotent stem cell therapy?
  4. Are there any mentionable safety aspects?
  5. How is the transplant material composed? How comparable is the transplant material to adult human islets?
  6. How will the persisting autoimmunity in T1DM affect the therapy goals?

In order to bring this review in line with the selected title, these points must be adressed properly. Figure 3 still misses an explanation of the "series of molecular signals".

Author Response

  1. Which type of patient (type 1, 2, age, gender, medical history, medical condition) is the target for a pluripotent stem cell therapy and who no ?.

  Thank you very much for your comment. According to your question, we attach the following paragraphs to provide an answer. However, we clarify that the purpose of the review article is not to focus on the medical conditions that the patients should have for iPSCs therapy.

Multiple studies in b-cell generation have been developed to date and, most of them have been focused on Type 1 diabetes mellites where the b-cells obtained have been derived from both healthy and diabetic individuals. Millman et al (2016), demonstrated that there was no difference between b-cells obtained from diabetic and non-diabetic patients, for the transplantation of b-cells in type 1 diabetic patients. However, an increasing number of subjects with type 2 diabetes mellitus have a complete deterioration of the function of the b-cells and require insulin therapy. Some new studies have shown that the effect of stem cell therapy may improve b-cell function in type 2 diabetic patients. However currently, stem cell therapy is indicated for young type 1 diabetic patients complicated by impaired awareness of hypoglycemia and recurrent severe hypoglycemia, extreme glycemic lability. In general, diabetic patients with “brittle” diabetes.

Some of the criteria that have been taken into account to exclude patients in clinical trials are a-) Body Mass Index (BMI) > 33; b-) Insulin requirements > 1.2 units/kg/day; c-) Significant kidney dysfunction. However, patients who have already had a kidney transplant may be candidates for the therapeutic strategy of pluripotent stem cells; d-) Significant liver/gall bladder disease; e-) Significant cardiovascular disease; f-) Active proliferative retinopathy,  Other clinical circumstances that proscribe the therapy with Pluripotent Stem Cells are High blood pressure despite appropriate treatment, High cholesterol/ triglycerides despite appropriate treatment, anemia or other blood disorders that require medical treatment, increased risk of bleeding, other chronic hemostasis disorders, or treatment with chronic anticoagulant therapy. Recent unresolved acute infection (except for mild skin infection or nail fungal infection), or chronic infection, Epstein-Barr Virus (EBV) IgG negative, any history of malignancy, except for completely resected squamous or basal cell carcinoma of the skin or in situ cancer of the cervix.  Recent history of non-adherence to medical treatment, or inability to demonstrate the capacity to comply with strict blood glycemic control and insulin therapy. Psychiatric illness that is untreated, or likely to interfere significantly with study compliance despite treatment. Presence of a chronic disease that must be chronically treated with a contraindicated agent.

2. Why is the cell implant being encapsulated in the current clinical study by Viacyte? One reason is to reduce the requirement for chronic immunosuppression but their is another one.

Thank you for clarifying this point. All of the devices, in Viacyte protocol, are designed to be subcutaneously implanted and allow oxygen, nutrients, proteins, and other molecules necessary for cell survival and function to reach the cells, and for insulin and other proteins secreted by the cells to leave the device.  In Viacyte’s protocols, the PEP-Encap device has generally prevented immune rejection and effectively protects implanted cells from the patient’s adaptive immune system. PEC-Direct and PEC-QT devices have the ability to stimulate neovascularization and are designed to allow blood vessels to enter the device and directly interact with the implanted cells. The cells implanted by the ViaCyte company are pancreatic precursor cells (PEC-01 cells). Once implanted under the skin of a patient, PEC-01 cells, which are contained within an implantation device, are designed to mature into functional b-cells and other islet cells that control the blood glucose levels. Because the cells are contained within and remain in the devices, they allow for retrieval of the implanted cells, an important safety consideration. We consider that the clinical use of pluripotent stem cells may take almost a decade for their practical application. Various groups around the world are working in this field, we consider in three main fields, the production of encapsulation devices, the development and optimal differentiation of pluripotent stem cells, and the editing of genes. Many of them with the support of pharmaceutical companies dedicated to the production of insulin such as Eli Lilly, Novo Nordisk.

3. What do we know about the current limitations of the pluripotent stem cell therapy?

Thank you very much for your question, this consideration gives us an overview of the harsh reality and arduous path that pluripotent stem cell research must follow. Perhaps, the main limitation found years ago with the use of ESCs was their ethical complications. However, the discovery of iPSCs put partially aside this inconvenience and allowed the advance of regenerative research in general limitations of pluripotent stem cell therapy is currently.

  • Stem cell-derived beta cells do not have a robust physiologic function.
  • Stem cell-derived islet-like clusters do not contain the full complement of endocrine cells.
  • More efficient low-cost differentiation methods are needed.
  • An effective delivery device that can support physiologic function and prevent immune attack has yet to be defined.
  • The optimal site for implantation has not been established.
  • Better T1D autoimmunity experimental models are needed for testing human cells.
  • Understanding which genetic modifications and therapies could allow cells to evade immune destruction, suppress immune responses, enhance functionality or better sustain the viability of cells after transplantation is vital to the future development of this technology.
  1. Are there any mentionable safety aspects?

Thanks again, precautions about the safety of pluripotent stem cell transplantation are derived from the risk of germline tumors such as teratomas. As mentioned in the article, safety studies are under development and it will be necessary to obtain the information from these studies to know some aspects of the safety and efficacy of these transplants with pluripotent stem cells in diabetic patients.

5. How is the transplant material composed ? How comparable is the transplant material to adult human islets ?

    In different studies, the material for b-cell transplantation has varied. In the case of Viacyte, they use pancreatic precursor cells that are derived from a single line of ESC, CyT49, looking for in vivo maturation after implantation. Other groups have considered pluripotent cells to be transplanted after stage (6) of differentiation. Other groups prefer the use of pancreatic islet clusters. In general, it is considered that the material for transplantation must be immature and that it must reach the functional characteristics of mature adult cells through differentiation after transplantation.

6.  How will the persisting autoimmunity in T1DM affect the therapy goals?

Thanks again. Although today the obtention of insulin-producing cells derived from iPSCs is a fact and can be considered as a mechanism for the disease treatment, it is necessary to fully understand the immunopathogenesis of type 1 diabetes mellitus to achieve the effective design of cell replacement therapies, as well as immunomodulatory strategies that allow not only to cure type 1 diabetes mellitus but also to reverse the autoimmunity that causes continuous graft rejection and restores the disease pathology. The subcutaneous implantation of the cells may allow periodic turnover if the implanted b-cells do not escape the recognition of the immune system of patients with type 1 diabetes mellitus. One possibility to avoid multiple transplants in the same individual is to edit the genes that induce the response and eventually create cell lines with these characteristics for allogenic transplantation. In fact, ViaCyte’s is developing a program that might answer the dual problems of low functionality and immune rejection through gene-editing technology being developed in collaboration with CRISPR Therapeutics. The clinical candidate PEC-QT is being developed as a gene-edited, immune-evasive cell line that would provide the functionality of insulin-producing cells while protecting the cells from a patient’s immune system.

 Figure 3 still misses an explanation of the "series of molecular signals.

An update of figure 3 was performed.

Round 3

Reviewer 2 Report

I have no further objection to acceptance of the manuscript.